# Cancer during Pregnancy: A Review of Preclinical and Clinical Transplacental Transfer of Anticancer Agents

**DOI:** 10.3390/cancers13061238

**Published:** 2021-03-11

**Authors:** Laure Benoit, Olivier Mir, François Vialard, Paul Berveiller

**Affiliations:** 1Centre Hospitalier Intercommunal de Poissy Saint-Germain-en-Laye, Department of Gynecology and Obstetrics, 78300 Poissy, France; laure.benoit@aphp.fr; 2Department of Ambulatory Cancer Care, Gustave Roussy, 94800 Villejuif, France; olivier.mir@gustaveroussy.fr; 3Université Paris-Saclay, UMR 1198, INRAE, BREED, RHuMA, 78350 Jouy-en-Josas, France; francois.vialard@uvsq.fr; 4Ecole Nationale Vétérinaire d’Alfort, BREED, 94700 Maisons-Alfort, France; 5Centre Hospitalier Intercommunal de Poissy Saint-Germain-en-Laye, Department of Genetics, 78300 Poissy, France

**Keywords:** pregnancy, cancer, placenta, anticancer agent, transplacental transfer

## Abstract

**Simple Summary:**

The use of anti-cancer treatments in pregnant women is an increasingly common situation given the increasing age at first pregnancy. The aim of this study is to review data concerning the transplacental transfer of drugs that can be used in the management of the most common pregnancy-associated cancers. This work is intended to guide clinicians in the choice of the treatments that offer the best benefit–risk ratio for the mother and the fetus and to deliver balanced information to pregnant patients.

**Abstract:**

The occurrence of cancer during pregnancy is observed in 1 in 1000 pregnancies and is expected to increase given the trend of delaying childbearing. While breast cancer is the most common, the incidence of other cancers, such as cervical, ovarian, and lung cancers as well as hemopathies and melanomas, is also increasing. Thus, cancer occurrence in pregnant women raises questions of management during pregnancy and, especially, assessment of the treatment benefit–risk ratio to ensure optimal management for the mother while ensuring the safety of the fetus. Chemotherapy remains a cornerstone of cancer management. If the use of anticancer agents appears possible during pregnancy, while avoiding the first trimester, the extent of placental transfer of different anticancer agents varies considerably thereafter. Furthermore, the significant physiological pharmacokinetic variations observed in pregnant women may have an impact on the placental transfer of anticancer agents. Given the complexity of predicting placental transfer of anticancer agents, preclinical studies are therefore mandatory. The aim of this review was to provide updated data on in vivo and ex vivo transplacental transfer of anticancer agents used in the management of the most common pregnancy-associated cancers to better manage these highly complex cases.

## 1. Introduction

The concomitant occurrence of cancer and pregnancy is 1 in 1000 pregnancies [1,2,3,4]. This incidence is increasing in industrialized countries owing to the trend of delaying pregnancy [5]. The most common solid malignancies during pregnancy are breast cancer, gynecological cancer, gastrointestinal cancer, and melanomas [5,6]. The management of a pregnant woman with cancer requires a multidisciplinary approach that must consider the benefit–risk ratio for the mother and fetus. The main parameters that influence the choice of treatment are gestational term; type and stage of cancer; the possibility of transplacental transfer and risk of teratogenicity of the drug; and the patient’s opinion on the continuation of the pregnancy if the disease is diagnosed at an early term [7]. While the treatment basis is often chemotherapy, targeted therapies and immunotherapy are becoming increasingly important in the treatment of solid cancers [8].

Although all chemotherapeutic agents can theoretically cross the placental barrier, the extent of placental transfer varies considerably from one compound to another [9]. Historically, three major mechanisms of placental transfer have been described: Passive diffusion, facilitated diffusion, and active transport [9]. The main physicochemical properties that influence placental transfer of molecules include molecular weight, lipophilia, ionization at physiological pH, and plasma protein binding [10]. Generally, highly lipophilic, low-molecular-weight molecules that are not ionized at physiological pH and weakly bound to plasma proteins are likely to cross the placental barrier more easily [9,10]. Most anticancer agents fulfill these criteria and can therefore theoretically cross the placenta and reach the fetal circulation [11]. However, other factors influence the transplacental passage of molecules, especially anticancer agents. For instance, some anticancer agents are substrates of efflux proteins expressed by human trophoblasts, such as ABCB1 and MDR1 and breast cancer resistance protein (ABCG2, BCRP) [10]. These proteins protect the fetus by preventing the passage of some anticancer drugs [10], and the transporters are involved in resistance to chemotherapy when they are overexpressed on the surface of tumor cells [10]. In addition, variations in the metabolism of pregnant women may have an impact on pharmacokinetic parameters. Maternal plasma volume increases by almost 50% in the third trimester of pregnancy [9], which induces an increased distribution volume for water-soluble drugs. Moreover, the concentration of albumin decreases, which may increase levels of unbound drugs and thus exacerbate potential fetal toxicity [12]. In parallel, renal clearance and liver oxidative metabolism increase during pregnancy, and increased activity of cytochrome P450 isoform 3A4 is also observed [13], which potentially leads to reduced maternal exposure to drugs metabolized by this isoenzyme.

Given the ethical considerations that make it difficult to conduct clinical trials in this setting, in vivo and ex vivo studies are required to assess drugs’ transplacental transfer. Regarding in vivo studies, animal models are not widely used because of differences in placentation and gestation [14]. Instead, the human fetal/maternal blood drug concentration ratio with a cord blood sample at delivery is usually used as a surrogate for drug transfer to the fetus [10]. Drug levels can also be assessed in amniotic fluid [15]. To determine the transport rate of xenobiotics across the placenta, preclinical data from ex vivo studies are crucial for completing the picture. The human perfused cotyledon model described by Schneider et al. [16] remains the “gold standard” for assessing the placental passage of drugs. Nevertheless, clinical data on the use of anticancer drugs in human pregnancy are scarce, heterogeneous, and are mostly represented by case reports.

In view of the paucity of data on chemotherapy, targeted therapies, and immunotherapy during pregnancy, we aimed to provide a summary of the available evidence from recent literature regarding the transplacental transfer of anticancer agents used in the management of the most common pregnancy-associated cancers, using data from in vivo and ex vivo studies. We focused on the most frequent solid tumors observed during pregnancy and excluded hematological diseases. This work is intended to guide clinicians in providing accurate information to patients and help them decide on the treatment that offers the best benefit–risk ratio for the mother and fetus.

## 2. Materials and Methods

### 2.1. Search Strategy

First, we reviewed the drugs used for the treatment of breast, gynecological, gastrointestinal, brain, and lung cancers as well as melanomas and sarcomas from the latest European Society for Medical Oncology (ESMO) guidelines [17] and the third consensus of the International Network on Cancer Infertility and Pregnancy (INCIP) [5]. Second, we carried out a systematic review of the literature by searching the PubMed and Web of Knowledge databases and the bibliography sections of relevant publications. The search terms were “drug name AND pregnancy,” “drug name AND placenta,” “drug name AND human perfused cotyledon,” “drug name AND transplacental transfer,” “drug name AND placental passage,” “drug name AND amniotic fluid,” and “drug name AND cord blood”.

### 2.2. Inclusion Criteria

We only included papers published in English and French between 1 January 1972 and 25 November 2020.

## 3. Results

### 3.1. Recent Guidelines on the Use of Anticancer Agents for the Treatment of Solid Tumors Most Often Associated with Pregnancy

#### 3.1.1. Breast Cancer

Breast cancer is the most common cancer during pregnancy, accounting for 40% of all types of cancers, with an incidence of 1 per 3000–10,000 pregnancies [5,6]. The diagnosis may be delayed or made at advanced stages due to pregnancy-induced physiological changes [5]. Prognosis of pregnancy-associated breast cancer appears to be the same as that for non-pregnant women, at identical ages and stages [18,19]. Disease-free survival and overall survival seems comparable for pregnant and non-pregnant women with a reported mortality rate of 14% and 12% respectively, during a median follow-up of 61 months [18], indicating that breast cancer prognosis is not affected by pregnancy. Indeed, a recent cohort of 65 patients with breast cancer during pregnancy did not show worse outcomes compared with that of 135 non-pregnant patients with breast cancer [20].

Anthracycline-based regimens (doxorubicin or epirubicin) remain the first choice, as recommended by the European Society for Medical Oncology (ESMO) clinical practice guidelines [17] and the International Network on Cancer Infertility and Pregnancy (INCIP) consensus guidelines [5]. Cyclophosphamide and taxanes can be added in sequence if needed, favoring the use of paclitaxel over docetaxel; however, the effects of fluorouracil (5-FU) remain uncertain [17]. For hormone-sensitive breast cancers, the use of tamoxifen is contraindicated during pregnancy because of its supposed association with fetal malformation [21]. HER2-targeted therapies (trastuzumab, pertuzumab) are also contraindicated as they expose the fetus to kidney damage and there is a high risk of oligo-anhydramnios [22].

#### 3.1.2. Cervical Cancer

Cervical cancer is the third most common cancer during pregnancy after breast cancer and lymphoma [5,6]. The incidence of cervical cancer ranges between 1 and 10 in 10,000 pregnancies [7,23] and therefore represents 13% of pregnancy-associated cancers [5,6]. Cervical cancer is usually diagnosed at an early stage during pregnancy, with a median gestational stage of 18.4 weeks of gestation (WG) [24], and the prognosis appears similar to that of non-pregnant patients [24,25]. Management of cervical cancer mainly depends on four criteria: Extent of local spread (tumor stage and tumor size), nodal status, term of pregnancy, and histological subtype [23].

In local stage IB1 with lymph node invasion and stages IB2 to IVA, the reference treatment is neoadjuvant chemotherapy. The ESMO clinical practice guidelines recommend the use of platinum-based chemotherapy with or without paclitaxel for the second trimester [11,17]. During the first trimester, if the patient wishes to preserve the pregnancy, close monitoring to delay treatment after 14 weeks is discussed [11]. In the third trimester, the treatment can be delayed until after delivery [17]. A cesarean section must be performed to avoid dissemination in the episiotomy or vaginal injury site, and a corporeal incision is advised to avoid abdominal dissemination [5]. The use of platinum-based chemotherapy during pregnancy is widely described in the literature, and its efficacy combined with good maternal and neonatal tolerance at birth and during the follow-up period (median: 17 months) make their use reasonable for pregnant women [26,27].

Regarding recent recommendations, carboplatin is now preferred over cisplatin due to the risk of dose-dependent ototoxicity in children after cisplatin exposure during pregnancy and the better maternal toxicity profile of carboplatin [11,28,29,30]. Whether area under the curve-based dosing formulas (Calvert or Chatelut) can be used for pregnant patients is unknown, and some clinicians use body surface area-based dosing in this setting [11].

#### 3.1.3. Ovarian Cancer

Ovarian cancer incidence is 1 in 10,000 pregnancies [7] and represents 7% of pregnancy-associated cancers [5,6]. Since a high frequency of non-epithelial tumors (e.g., germ cell or sex cord stromal tumors) is reported due to the young age of patients [5], we only focused on the treatment of non-epithelial ovarian cancers.

The ESMO clinical practice guidelines recommend adjuvant chemotherapy with the same indications as for non-pregnant patients for whom the combination of bleomycin-etoposide-cisplatin (BEP) or etoposide-cisplatin (EP) is usually used [30]. While previous recommendations propose vinblastine as an alternative to bleomycin or the association of paclitaxel and carboplatin during pregnancy, the latest guidelines still recommend BEP or EP regimens for treating non-epithelial ovarian cancers [11].

Myelotoxicity with secondary leukemia is a well-known side effect of etoposide. However, its use during pregnancy in association with cisplatin with or without bleomycin has been described and appears to be safe [11], although the number of cases is limited [31,32,33].

#### 3.1.4. Melanoma

With an incidence ranging from 1 to 2.6 in 10,000 pregnancies [7], melanoma represents 5% of pregnancy-associated cancers [5,6]. The overall incidence has been increasing over the last couple of years, particularly in premenopausal women [34]. Diagnosis is most often made at an advanced stage, possibly due to delayed diagnoses attributable to the tendency for both patients and health workers to address changes in pigmentation as only physiological due to pregnancy. Compared with non-pregnant patients, pregnant women with melanoma do not appear to have a poorer prognosis at identical ages and stages [34,35]. The ESMO clinical practice guidelines recommend identical surgical treatment as for non-pregnant patients [17], and there is no recommendation for systemic treatment during pregnancy. Systemic treatment for non-pregnant patients relies on immunotherapy with immune checkpoint inhibitors targeting PD-1 and/or CTLA4 and protein kinase inhibitors targeting BRAF and MEK [36]. Dacarbazine and interferon alpha are now considered second- or third-line therapies [35]. The last ESMO clinical practice guidelines mention that ipilimumab (a monoclonal antibody against CTLA4) or vemurafenib (a BRAF inhibitor) should not be used during pregnancy because of the lack of safety data, proposing an alternative to interferon-alpha [17]. Even though contradictory findings have been reported for BRAF inhibitors [37,38,39,40], the use of immune checkpoint inhibitors appears relatively safe based on several reports [41,42,43,44].

#### 3.1.5. Gastrointestinal Cancers

Gastrointestinal cancers represent 4–5% of pregnancy-associated cancers [5,6], with the incidence of colorectal cancer being 1 in 13,000 pregnancies [7]. The standard chemotherapy for advanced colorectal carcinoma is based on FOLFOX or FOLFIRI regimens (5-FU, leucovorin, and oxaliplatin or irinotecan) [45,46,47]. There are no specific guidelines for the management of colorectal cancer during pregnancy, but studies consider the FOLFOX regimen feasible during the second and third trimesters [48]. Indeed, 5-FU is considered to not result in significant long-term disabilities, apart from the tendency for newborns to be smaller than those who are not exposed, and oxaliplatin and irinotecan have also been reported to be safe [47,48,49,50,51,52]. The US Food and Drug Administration has approved targeted therapies as first-line treatment for patients with metastatic colorectal cancers in combination with 5-FU-based chemotherapy [47], such as bevacizumab (a monoclonal antibody against VEGF) and cetuximab (a monoclonal antibody against EGFR), but limited data have discouraged their use during pregnancy.

#### 3.1.6. Brain Cancer

In non-pregnant patients with glioblastoma, temozolomide remains the standard of care [53]. Standard therapy for newly diagnosed anaplastic astrocytoma and anaplastic oligodendroglioma includes neoadjuvant or adjuvant chemotherapy with a procarbazine–lomustine–vincristine regimen [53]. Because of their rarity (1–2% of pregnancy-associated cancers) [5,54], there are no guidelines for the adjuvant systemic treatment of brain malignant tumors during pregnancy. Given the paucity of data in the literature concerning the use of chemotherapy for their treatment during pregnancy, the only available information is from cases reporting the discovery of pregnancy in a patient treated for brain cancer [55]. In such a case, the use of alkylator-based chemotherapy (temozolomide or PCV) results in the birth of a healthy infant [55].

#### 3.1.7. Other Cancers

##### Non-Small Cell Lung Cancer

Non-small cell lung cancer (NSCLC) accounts for 85% of lung cancers, and although its diagnosis during pregnancy is rare (less than 1% of pregnancy-associated cancers) [5], it is increasing in pregnant women and often occurs at an advanced stage in 98% of cases [17]. Oncogene mutations are more frequent among cases of NSCLC that occur during pregnancy [56]. In cases of NSCLC with EGFR activating mutations, the protein kinase inhibitors gefitinib, erlotinib, and osimertinib are standard first-line treatments [57,58,59]. NSCLC with anaplastic lymphoma kinase (ALK) rearrangement is less frequent than NSCLC with EGFR mutations, and is usually treated with crizotinib or alectinib [56]. The ESMO clinical practice guidelines recommend the combination of carboplatin and paclitaxel for the treatment of NSCLC during pregnancy, but discourage the use of protein kinase inhibitors given the current lack of data [17]. Regarding small cell lung cancer, the standard EP regimen is feasible (30). As mentioned earlier for melanomas, the use of immune checkpoint inhibitors during pregnancy has scarcely been documented [41,42,43,44].

##### Soft Tissue Sarcoma

For patients with soft tissue sarcoma who require neoadjuvant or first-line chemotherapy, the ESMO clinical practice guidelines recommend the use of doxorubicin. The combination of doxorubicin and ifosfamide during pregnancy has been reported, but data are limited to a few cases [60] and hence the toxicity profile of ifosfamide cannot be confirmed; thus, doxorubicin remains recommended as a monotherapy in this case.

### 3.2. Preclinical Data on the Placental Transfer of Anticancer Agents

Data regarding transplacental transfer of drugs are summarized in Table 1. Only studies documenting fetal transfer of the drugs are mentioned (in vivo data correspond to the dosage of the drug in cord blood, amniotic fluid, or fetal tissues; and ex vivo data correspond to the fetal transfer rate of a drug using a human perfused cotyledon model).

#### 3.2.1. Antimetabolite Agents: Fluorouracil

As previously mentioned, 5-FU-based chemotherapy is used for the treatment of gastrointestinal cancers and breast cancer. However, no study assessing the transplacental transfer of these drugs in humans has been published to date. We have only found one study on the transplacental transfer of 5-FU in a rat model [61]; transplacental transfer was assessed using maternal and fetal plasma samples after 5-FU administration to pregnant rats. A significant amount of 5-FU crossed the placenta and the relative fetal exposure increased in a dose-dependent manner (17.8%, 28.7%, and 52.3% at 10, 25, and 100 mg/kg doses, respectively).

#### 3.2.2. Antimitotic Agents

##### Vinca Alkaloids: Vincristine, Vinblastine

In vitro studies have demonstrated placental transfer of vinblastine and vincristine. One research group used brush border membrane vesicles from human trophoblast cells to demonstrate the crucial role of ABCB1 in vinca-alkaloid uptake [95]; the authors confirmed that administration of various ABCB1 inhibitors leads to significant reductions in the uptake of vinblastine and vincristine [95]. Moreover, a recent study described the effects of vinblastine on human cytotrophoblasts [96], which included decreased cell viability. Another in vitro study demonstrated the transfer of vinblastine through layers of cultured cells or brush border membrane vesicles [97]. This indicates that ABCB1 is present in placental membrane vesicles and that vinblastine is an ABCB1 substrate. Indeed, administration of an ABCB1 inhibitor significantly reduces the vinblastine uptake.

Notably, one human ex vivo study with human perfused placenta confirmed the role of ABCB1 inhibition on the transplacental transfer of vinblastine, and the vinblastine clearance index is significantly higher in the presence of an ABCB1 inhibitor (0.34. vs. 0.25 in its absence) [98].

We did not find any published data on the in vivo maternal pharmacokinetics or transplacental transfer of vinblastine and vincristine in humans. There were, however, two studies on the transplacental transfer of vinblastine in animal models [62,63]. In the first study, pregnant mice were injected with vinblastine [62], and the fetal concentration of the drug reached 14% of the maternal concentration. In a baboon model, the fetal concentration was found to be 18.5% of the maternal concentration, but the compound was not detected in the fetal brain or cerebrospinal fluid [63].

##### Taxanes: Paclitaxel, Docetaxel

A recent in vitro study on human cell lines showed that paclitaxel moderately affects explant viability and reduces cell viability by 50% or more in trophoblast cell lines [99]. The same results were reported with docetaxel, with a maximum 30% decrease in cell viability [96]. Nevertheless, the use of taxanes during pregnancy is widely described. Several studies have reviewed fetal and neonatal outcomes after intrauterine exposure in pregnant women who underwent taxane-based chemotherapy for cervical [100], ovarian [101,102,103,104] and breast cancers [104,105,106,107,108]. According to these reports, taxanes appear to have a favorable toxicity profile. In vitro, in vivo, and ex vivo studies support these data.

Four human ex vivo studies assessed the transplacental transfer of taxanes with the human perfused placental lobule model. A study investigating the transplacental transfer of paclitaxel [65] demonstrated a fetal transfer rate of 4.3% (n = 7 placentas). Paclitaxel was shown to be a substrate of P-glycoprotein in in vitro studies with human trophoblast cell lines [66]. Accordingly, the authors examined its transfer rate in the presence of ABCB1 inhibitors and observed transfer rates reaching 8.8% (n = 6 placentas), confirming the role of ABCB1 in protecting the fetus against drugs such as paclitaxel. Similar findings were reported with the same model (n = 12 placentas) [67] as the fetal transfer rate of paclitaxel was found to be 3.97% and 6.56% in the presence and absence of ABCB1 inhibitors, respectively. Furthermore, transplacental transfer of paclitaxel (n = 3 placentas) and docetaxel (n = 3 placentas) were compared [68] and demonstrated a low placental transfer rate (1.72% and 4%, respectively), although there was high inter-patient variability. Taxane accumulation in cotyledons was about 3–4%, which is similar to that of docetaxel and paclitaxel. More recently, the fetal transfer rate of paclitaxel was found to reach 7% in one term placenta [69], confirming inter-placental variability.

One study provided crucial data on the maternal pharmacokinetics of taxanes in pregnant women [109]. Interestingly, the authors observed suboptimal exposure to paclitaxel and docetaxel during pregnancy, and thus suggested that higher doses of taxanes may be required for optimal effectiveness; the calculated dose adjustment requirements were +37.8% and +16.9% for paclitaxel and docetaxel, respectively. Similar findings have been reported in a rat model [110].

In animal studies, mouse, rat, and baboon models were used to study the transplacental transfer of taxanes [62,64]. Regarding paclitaxel, there is no evidence of placental transfer in the mouse model [62] but paclitaxel is detected in fetal tissues in the baboon model [64]. Docetaxel in the baboon model is not detected in fetal blood samples but is detected in fetal tissues 3 h after docetaxel infusion. Fetal tissues contain 5–50% of maternal tissue concentrations, and levels were similar at 3, 26, or 76 h after administration [64]. In the rat model, fetal uptake of paclitaxel is markedly lower than that of the placenta [110].

In vivo transplacental transfer of taxanes in humans has been documented in only one publication [70]. The authors collected meconium samples from 23 newborns whose mothers underwent taxane-based chemotherapy during the second or third trimesters. The mean levels of paclitaxel (399.9 pg/mg) and its metabolites in eight screened samples were assessed, confirming human fetal exposure. The authors concluded that variability in meconium levels between individuals may indicate a potential for reducing fetal exposure based on timing, dosing, and individual characteristics [70].

To conclude, in light of these data, the use of taxanes in pregnant women seems to be feasible during the second and third trimesters of pregnancy.

#### 3.2.3. Alkylating Agents

##### Cyclophosphamide

Consequences of prenatal exposure to cyclophosphamide were reviewed in a recent study [111]. Side effects after second or third trimester exposure involved intrauterine growth restriction; this was confirmed in a cohort of 24 fetuses exposed to cyclophosphamide [105]. However, another study concluded that prenatal exposure to cyclophosphamide-based chemotherapy does not impact fetal brain growth [112].

We did not find any ex vivo studies on the transplacental transfer of cyclophosphamide.

Cyclophosphamide placental transfer has been documented in an animal in vivo study, where limited transplacental transfer of cyclophosphamide was reported in a baboon model [63]; 4-hydroxy-cyclophosphamide concentrations in fetal plasma and cerebrospinal fluid averaged 25% and 63% of the maternal concentrations, respectively.

In humans, there are limited data on the capacity of cyclophosphamide to cross the placenta. To date, only one human study has documented the transplacental transfer of cyclophosphamide in vivo [71]. A 33-year-old pregnant woman diagnosed with Hodgkin’s lymphoma was treated with cyclophosphamide from 29 WG. The level of cyclophosphamide in amniotic fluid collected via amniocentesis at 34 weeks, 1 h after the last injection, was equivalent to one quarter of the maternal plasma level.

##### Platinum Derivatives: Cisplatin, Carboplatin, and Oxaliplatin

To our knowledge, no study has focused on the transplacental transfer of oxaliplatin, and only case reports of FOLFOX administration during pregnancy have been described.

An in vitro study on human placental tissue explants and trophoblast cell lines exposed to cisplatin or carboplatin concluded that both drugs affect cell and tissue viability at clinically relevant concentrations [99]. Hence, more data on placental transfer are required.

To our knowledge, only three studies have reported placental transfer of platinum salt with the human placental perfusion model [75,78,79]. The only study on cisplatin placental transfer reported that cisplatin transport is negligible in the human placenta at term (13% of the value for antipyrine, the reference marker) [75]. The same group used the human ex vivo model to assess placental transfer of carboplatin and found that the transport fraction also averaged 13% of the antipyrine transfer rate [78]. Thus, the authors suggested that there is minimal fetal risk when using cisplatin or carboplatin for pregnant patients. However, the results of these two studies should be interpreted with caution given the very short perfusion time of 5 min. Another group used the same ex vivo model and found that transplacental transfer of carboplatin is concentration-dependent [79]. Hence, the authors suggested that it may not be necessary to empirically reduce carboplatin doses for pregnant women.

Studies in animal models have also demonstrated placental transfer of platinum salts. One study in pregnant mice described the transplacental transfer of labeled cisplatin at different points during gestation [72]. Interestingly, very small amounts of radioactivity were detected in the embryos during the first few days of gestation, which then increased during later stages of gestation. These results suggest that placental transfer may depend on gestational age, and that placental maturation and a progressive increase in transporter expression may influence drug transfer. Other studies confirmed the transplacental transfer of carboplatin in mice with a fetal transfer rate up to 117% [62], and in baboons, with fetal concentrations of up to 57.5% of the maternal concentration [63], a rate clearly higher than the fetal transfer rate of 13% previously mentioned reported with the human perfused cotyledon [78]. The capacity of cisplatin to cross the placental barrier has also been shown in the patas monkey [73]. Finally, one study focused on the accumulation of cisplatin versus cisplatin associated with a bile acid moiety, glycocholic acid, in fetal tissues in rats [74]. The objective was to elucidate whether the coupling of cisplatin to a the glycocholic acid could endow beneficial properties; namely, the ability of the placenta to prevent the passage of the drug toward the fetal compartment. The authors demonstrated a higher cotyledon accumulation of cisplatin when non-associated with glycocholic acid, suggesting an alternative drug in the treatment of certain tumors during pregnancy.

In humans, several reports of use of platinum derivatives have been reported. One study reported severe bilateral perceptive hearing loss in a neonate following intrauterine exposure to cisplatin [28]. However, a large amount of data supports the safety of platinum-based chemotherapy during the second and third trimesters. A systematic review found 43 cases of pregnant women treated with platinum derivatives [27]; thirty-six patients received cisplatin, six received carboplatin, and one received both drugs. Two fetal malformations occurred, but the role of cisplatin remains questionable because of the short delay between malformation occurrence and cisplatin administration. These data are in agreement with the results of another cohort where cisplatin concentrations in the umbilical cord and amniotic fluid were reported to be 31–65% and 13–42% of the amount in maternal blood, respectively [76]. Another study reviewed 47 pregnancies from 24 studies, in which pregnant patients underwent platinum-based chemotherapy [113]. All children were healthy, with a median follow-up of 12.5 months. Finally, a recent meta-analysis including 88 patients from 39 studies documented asymptomatic transplacental transfer of platinum-based chemotherapy during the second and third trimesters [26].

Several human in vivo studies have demonstrated significant placental transfer of cisplatin with cord blood samples. In one study, neonatal cisplatin levels were 40 µg/mL on day 3 post-chemotherapy [80]. In another, cisplatin levels were 0.82 µm/L in cord blood and 1.10 µm/L in maternal plasma 2 weeks after the last cycle of chemotherapy [81]. Moreover, in a cohort of seven patients with cervical cancer treated with neoadjuvant cisplatin during pregnancy, all patients delivered healthy babies (mean follow-up of 7 months) [77]. Cisplatin concentrations in umbilical cord blood and amniotic fluid were found to be 31–65% and 13–42% of that in maternal blood, respectively. Cisplatin was also measured in amniotic fluid collected via amniocentesis [82,83]. Furthermore, a study described the case of a 35-year-old patient with a twin pregnancy treated with cisplatin for cervical cancer [15]. The cisplatin concentration in amniotic fluid samples was 106.7 mg/L and reached 10% of the maternal blood levels. At delivery, cisplatin concentrations in the cord blood of twin neonates were 57.1 and 61.2 mg/L. The corresponding concentration in the amniotic fluid was approximately one-third of the umbilical cord. The twins developed normally and displayed no chemotherapy-related side effects. Another study reported the detection of platinum-DNA adducts in cord blood lymphocytes 9 weeks after the last injection of carboplatin administered during 22 WG. A cohort of 21 patients receiving platinum salts during pregnancy delivered 22 healthy babies without renal, hepatic, auditory, or hematopoietic impairment (follow-up period between 7 and 88 months) [114]. Platinum concentrations in umbilical cord blood and amniotic fluid were 23–65% and 11–42% of the maternal blood, respectively.

To conclude, the benefit–risk ratio for mother and fetus seems to be in favor of the use of platinum derivatives during the second and third trimesters of pregnancy, by giving priority to carboplatin due to its favorable fetal toxicity profile.

##### Dacarbazine

To date, there have been no ex vivo nor in vivo human studies on the placental transfer of dacarbazine in the literature. One pharmacokinetic study in humans suggested that pregnancy appears to decrease metabolism of the pro-drug dacarbazine [115].

#### 3.2.4. Topoisomerase II Inhibitors

##### Anthracyclines: Doxorubicin, Epirubicin

An in vitro study on human placental tissue explants and trophoblast cell lines exposed to doxorubicin concluded that it affects both cell and tissue viability at clinically relevant concentrations [99]. Similarly, a decrease in human cytotrophoblasts viability induced by epirubicin has been demonstrated in vitro [96].

In humans, a pharmacokinetic study concluded that there is suboptimal exposure to anthracyclines during pregnancy due to gestation-induced changes, and suggested that the dose should be increased by 5.5% and 8% for doxorubicin and epirubicin, respectively [109]. A recent study on the pathways involved in anthracycline-induced gestational complications (preeclampsia and IUGR) [116] used both a mouse model and in vitro human placental cells to demonstrate the direct toxicity of doxorubicin on trophoblasts.

One ex vivo study assessed the transplacental transfer of liposomal formulations of doxorubicin (non-PEGylated and PEGylated liposomal doxorubicin) with the human perfused cotyledon model [85]. Interestingly, non-PEGylated liposomal doxorubicin crossed the placental barrier (reaching 12% of the peak maternal concentration), but PEGylated liposomal doxorubicin did not. Only one old French study reported non-modified doxorubicin placental transfer in an ex vivo model [117]. Recently, Shah et al. [118] developed an analytical method of high performance liquid chromatography coupled with fluorescence detection to determine the concentration of doxorubicin in cell culture media for transport studies in human trophoblast cells and fetal media for placental perfusion. Studies on transplacental transfer of doxorubicin with the human perfused cotyledon model are therefore expected in the future.

Several studies have reported preclinical in vivo data regarding placental transfer of anthracyclines in animals [62,63]. Maternofetal transfer rate reported averaged 4% to 7.5% (Table 1). A study suggested that administration of doxorubicin to pregnant mice affects offspring’s brain development and behavior [119].

In humans, doxorubicin is known to have cardiotoxic effects. A case of neonatal cardiomyopathy attributable to a doxorubicin-based regimen has been reported [120]. The cardiotoxic effects of doxorubicin on neonatal hearts can be improved by physical exercise during pregnancy [121]. One study on the pharmacokinetics of doxorubicin in pregnant women indicated a decreased clearance of doxorubicin during pregnancy [122]. According to several case reports, the use of doxorubicin appears relatively safe during pregnancy [105,123,124,125,126]. In contrast, a case of intrauterine death after exposure to epirubicin for breast cancer has been reported [127]. One study assessing the impact of in utero exposure to doxorubicin on brain growth concluded that anthracycline-based chemotherapy does not affect fetal growth [112]. A review showed a similar toxicity profile for epirubicin and doxorubicin during pregnancy based on pharmacological evidence [128].

In a literature review, Germann et al. [129] collected studies on embryo-fetal outcomes in 160 patients who underwent anthracycline treatment during pregnancy between 1976 and 2001. The fetal and neonatal outcome was found to be normal in 73% of cases, with only 3% fetal malformations reported. The risk of severe fetal toxicity increased 30-fold when the doxorubicin dose per cycle exceeded 70 mg/m^2^. Other clinical data on the use of doxorubicin or epirubicin during pregnancy are only based on case reports. For instance, a case study of nine patients treated with doxorubicin in association with ifosfamide during pregnancy reported favorable outcomes for all mothers and offspring [60]. Interestingly, neonatal follow-up of a recent case of maternal chemotherapy with a doxorubicin-containing regimen in a dichorionic diamniotic pregnancy [86] revealed cardiac dysfunction in one twin, suggesting a role for specific fetal factors that confer variability in the pharmacokinetics of doxorubicin.

Matalon et al. [130] reviewed studies on the in vivo placental transfer of doxorubicin in humans. In a patient receiving doxorubicin for breast cancer at 25 WG, no doxorubicin was detected in the placenta and cord blood at delivery 3 weeks after the last treatment [87]. Moreover, doxorubicin and its metabolite were not detected in the amniotic fluid collected via amniocentesis from a patient treated with doxorubicin for breast cancer, 96 h after the last administration [88]. Another study reported two cases of pregnant women treated with doxorubicin for lymphomas [89]; both patients delivered shortly after the last administration of doxorubicin. The first patient gave birth to a healthy child, and doxorubicin was detected in the placenta but not in the cord blood. For the second woman, the child was stillborn. Notably, doxorubicin was not detected in fetal tissues but high levels of its metabolites were found in fetal organs [89]. Other studies [90,91] did not detect doxorubicin in the amniotic fluid but found high concentrations (10 times the maternal concentration) in fetal lungs, liver, and kidneys. The authors reasoned that the lack of doxorubicin in the amniotic fluid was due to the presence of a large distribution volume during pregnancy.

Regarding all these data, the placental transfer appears weak, suggesting a possible use during the second and third trimesters of pregnancy if urgent treatment is required.

##### Etoposide

No data regarding in vivo or ex vivo human placental transfer of etoposide have been reported in the literature. One in vivo study assessed etoposide toxicity on trophoblasts in mice [84], where injecting etoposide to pregnant mice was found to induce apoptosis and growth arrest of mouse placenta trophoblasts. The authors suggested that this phenomenon may induce severe intrauterine growth restriction.

#### 3.2.5. Topoisomerase I Inhibitors: Irinotecan

To our knowledge, there are no published data on the placental transfer of irinotecan in animal model or in humans.

#### 3.2.6. DNA-Cleaving Agent: Bleomycin

To our knowledge, no human in vitro, ex vivo, or in vivo studies of the maternofetal transfer of bleomycin have been published to date. A recent retrospective study on the use of bleomycin during pregnancy included 72 pregnant patients who underwent bleomycin chemotherapy (ABVD regimen) for Hodgkin’s lymphoma [124]; no fetal malformations were reported in this cohort.

#### 3.2.7. Therapeutic Monoclonal Antibodies: Bevacizumab, Cetuximab, Ipilimumab, Trastuzumab, Nivolumab, and Pembrolizumab

We did not find any preclinical or clinical data on placental transfer of cetuximab, ipilimumab, bevacizumab, or trastuzumab in humans. On the other hand, several recent case reports have described successful maternofetal outcomes after prenatal exposure to ipilimumab during pregnancy, and no congenital anomalies were observed [42,43,44]. In contrast, in pregnant monkeys, ipilimumab leads to a dose-dependent increase in miscarriages, stillbirths, and premature births, and there is a questionable association with malformations of the urogenital tract [131]. Indeed, as an IgG1, ipilimumab crosses the placenta.

Bevacizumab is expected to cross the placenta easily (another IgG1 antibody), the ease of which increases with advancing gestational age [132]. The only available studies on the use of bevacizumab in pregnant women involve intravitreal injection for managing diabetic retinopathy and choroidal neovascularization. Other data come from animal models; one study assessed transplacental transfer of bevacizumab in pregnant rats after injecting fluorescent-labeled bevacizumab at different gestational ages [92]. Bevacizumab was detected in the embryo as of gestation day 13 until the end of gestation in a dose-dependent manner and as early as 24 h after injection. Another team studied the effect of intraperitoneal injection of bevacizumab on early embryo development in pregnant rats and observed an inhibitory effect on pregnancy development and litter death [133]. Moreover, bevacizumab has been shown to have teratogenic effects in rabbits when administered at therapeutic human doses [134]. In addition, bevacizumab is known to induce a preeclampsia-like syndrome associated with inhibition of the VEGF pathway [135].

Transplacental transfer of trastuzumab was assessed in pregnant baboons [64] and was found to reach 85.0% 2 h after trastuzumab infusion. After 26 h, the placental transfer was 3% and the amniotic fluid contained 36.4% of the fetal plasma concentration [64]. In humans, the use of trastuzumab during pregnancy has been shown to be associated with anhydramnios [22,106,136,137] related to fetal renal insufficiency [138], and thus trastuzumab is currently not recommended during pregnancy [5].

To our knowledge, there are no published data on the placental transfer of recent PD-and PD-L1 inhibitors such as nivolumab and pembrolizumab in animal models or in humans.

#### 3.2.8. Protein Kinase Inhibitors: Gefitinib, Erlotinib, Osimertinib, and Vemurafenib

The use of protein kinase inhibitors during pregnancy is currently not recommended, given the limited available data [11].

A study showed that combining gefitinib with methotrexate potently inhibits placental cell growth in in vitro human cell lines and an in vivo mouse model [57]. Another in vitro study concluded that gefitinib and crizotinib significantly affect viability of human placental explants, trophoblast cell lines [99], and BeWo cells [94].

To our knowledge, only two studies have assessed the transplacental transfer of protein kinase inhibitors with the human perfused cotyledon model, including a French group that compared the transplacental passage of anti-EGFR drugs (n = 9 placentas) [57]. The authors showed a fetal transfer rate of 16.8% and 31.4% for gefitinib and erlotinib, respectively, and therefore suggested that gefitinib may be preferrable over erlotinib for administration to pregnant women. This low transplacental transfer of gefitinib has also been reported in an in vivo human study [93]. In a patient treated for EGFR-mutated cancer, gefitinib concentration was 25.7 ng/mL in cord blood and 1.9 ng/mL in amniotic fluid, which is 20% of the maternal concentration after delivery. The use of erlotinib or gefitinib during pregnancy, including after first trimester exposure, is not associated with fetal or neonatal malformation according to four case reports [93,139,140,141]. In another ex vivo study [94], the authors used the perfused human cotyledon model with crizotinib (n = 9 placentas), and showed a low transfer rate across the placenta. Data are lacking regarding osimertinib and alectinib, now recommended as first-line treatment for EGFR-mutated and ALK-rearranged NSCLC, respectively.

In humans, ALK-rearranged NSCLC during pregnancy has been described [56,142,143], and one study reported intrauterine growth restriction in one twin of a dichorionic diamniotic pregnancy after intrauterine exposure to erlotinib [144], but normal development at 12 months. These data should be interpreted with caution given the suspected toxicity of these drugs to trophoblasts.

Regarding the treatment of melanomas, there are no data regarding placental transfer of vemurafenib using the ex vivo model in humans. In animal studies employing rats and rabbits, vemurafenib has been shown to cross the placenta without displaying any teratogenic effects [131].

In humans, data on the use of vemurafenib during pregnancy are limited to case reports [37,39,145]. No fetal or neonatal malformations have been reported, but one case described intrauterine growth restriction that required a cesarean section at 30 WG. Blood samples from the mother, newborn, and umbilical cord were tested for the presence of vemurafenib and showed plasma values in the infant and cord blood (10.9 µg/mL) corresponding to half of the maternal value (24.3 µg/mL) [40]. Another study reported a case of severe toxic epidermal necrolysis after a patient with metastatic melanoma and a monochorionic-diamniotic twin pregnancy was treated with vemurafenib [39].

## 4. Conclusions

This review aimed to summarize the available data and evidence regarding the placental transfer of anticancer agents used for treating the most frequently occurring cancers during pregnancy. We chose not to study hematological diseases, as the treatment and management of acute leukemias and lymphomas are specific. Another limitation was the limited data on the transplacental transfer of anti-cancer drugs during pregnancy due to the relative rarity of their use and the lack of studies with large numbers of patients. Nevertheless, multi-disciplinary management must be performed on a case-by-case basis, taking into account the disease stage, term of the pregnancy, and wishes of the patient. Complementary clinical follow-up data on children with intrauterine exposure to anticancer agents are warranted to better guide physicians and provide more information to pregnant women diagnosed with cancer.

## Figures and Tables

**Table 1 cancers-13-01238-t001:** Overview of transplacental transfer of anticancer agents.

	EXPERIMENTAL MODELS		
ANTICANCER AGENTS	In Vivo Data on Animal Models	Ex Vivo Human Data (Human Perfused Cotyledon Model)	In Vivo Human Data
Antimetabolite agents			
Fluorouracil	Rat model: Related fetal exposure = 28.7% [61]	No data	No data
Antimitotiques agents			
Vinca alcaloïds			
Vinblastine	Mouse model: FTR = 13.8% [62] Baboon model: FTR = 18.5% [63]	No data	No data
Vincristine	No data	No data	No data
Taxanes			
Paclitaxel	Mouse model: No evidence of placental transfer [62] Baboon model: FTR = 1.5%, detectable in fetal tissues [64]	FTR = 1.72% to 7% [65,66,67,68,69]	Detected in neonate’s meconium (399.9 pg/mg) [70]
Docetaxel	Baboon model: Undectectbable in fetal blood, detected on fetal tissues [64]	FTR = 4% [68]	No data
Alkylating agents			
Cyclophosphamide	Baboon model: FTR = 25% [63]	No data	Detected in AF: 25% of maternal plasma level 1h after injection [71]
Dacarbazine	No data	No data	No data
Platinum derivatives			
Cisplatin	Mouse model: FTR gestational age-dependent [72] Pata monkeys/rats models: Detected in fetal and neonate’s tissues [73,74]	FTR = 9% [75]	Detected in cord blood (23–65% of MC), AF (10–42% of MC), and placental tissues [76,77]
Carboplatin	Mouse model: FTR = 117% [62] Baboon model: FTR = 57.5% [63]	FTR = 4–13% [78,79]	Detected in cord blood, AF, and placental tissues [15,77,80,81,82,83]
Oxaliplatin	No data	No data	No data
Topoisomerase inhibitors			
Etoposide	Mouse model: Induced apoptosis in trophoblasts [84]	No data	No data
Anthracyclines			
Doxorubicin	No data	FTR of non-pegylated liposomal doxorubicin = 12%/FTR of pegylated liposomal doxorubicin = 0% [85]	Undetectable in cord blood, AF, or placenta at delivery [86,87,88] Detectable in fetal organs [88,89,90,91]
Epirubicin	No data	No data	No data
Irinotecan	No data	No data	No data
Other DNA modifying agent			
Bleomycin	No data	No data	No data
Monoclonal antibodies			
Bevacizumab	Rat model: Detectable in the embryo at day 13 [92]	No data	No data
Cetuximab	No data	No data	No data
Ipilimumab	No data	No data	No data
Trastuzumab	Pregnant baboons: FTR = 85% after 2 h and detected in AF (36.4% of the fetal plasma concentration after 26h) [64]	No data	No data
Protein kinase inhibitors			
Gefitinib	No data	FTR = 16.8 % [57]	Detected in cord blood (25.7 ng/mL) and AF (16.9 ng/mL) = 20% of MC [93]
Erlotinib	No data	FTR = 31.4 % [57]	No data
Osimertinib	No data	No data	No data
Vemurafenib	No data	No data	Detected in cord blood (10.9 µg/mL) = 50% of MC [40]
Crizotinib	No data	FTR < 6% [94]	No data

Abbreviations: FTR = fetal transfer rate, AF = amniotic fluid, MC = maternal blood concentration.

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
