# Peer review of "Cancer during Pregnancy: A Review of Preclinical and Clinical Transplacental Transfer of Anticancer Agents"

_cancers, 2021, doi:10.3390/cancers13061238_

Round 1
Reviewer 1 Report
The authors provide a nice summary of the available vidence for the use of systemic cancer therapies during pregnancy. I've the following comments/suggesitons:
- Several other commonly used cancer drugs are missing in the review - e.g. imatinib, PD1/PD-L1 inhibitors etc. Please include.
- Irinotecan is disucssed in section 2.5, but not placental transfer data in section 3. Please include.
- Consistent terminology is recommended to be used throughout the manuscript- e.g. ABCB1 vs P-glycoprotein - these are used interchangably
- Table 1 is not included for review
- Line 228 - "a significant amount of 5-FU crossed the placenta and the relative fetal exposure increased in a dose-dependent manner" - Please include numbers for the "signifciant amount"
- Line 346 - FTR? - no expanded version of this
- Section 5.2 - line 569 - "between" is usually followed by "and" and not "to".
- Author contributions, funding, IRB statement, Informed consent statements - missing
Reviewer 2 Report
Dear Authors,
I have some suggestions for you how to improve your manuscript:
- English revison is needed.
- 4th line of introduction- if you write percents write them in all cases or in none.
- Part of ovarian cancer needs improvements. I cannot agree with opinion that treatment of epithelial and non-epithelial ovarian cancer is the same in both cases. That are two different things.
- Material and methods should be put in other part of the text; inclusion criteria are too poor if we talk about systematic review.
- If you write Material and methods the part called results is needed
- What about geographical correlation and melanomas?
Reviewer 3 Report
Benoit et al. summarised the knowledge of the effect of certain cancer treatments during pregnancy.
Table 1 and supplementary data are missing in the manuscript. These are essential components as the text referred to them.
Round 2
Reviewer 3 Report
Benoit et al. summarised the knowledge of the effect of certain cancer treatments during pregnancy. The table 1 summarizes the data found in during the search for writing the review. I believe that there are still missing information in the table.
For example papers 31 and 33 investigate the effect of Vinblastine, Cisplatin and bleomycin in pregnant women. In the table in the column under in vivo human data it is just mentioned no data for Vinblastine, Bleomycin and under cisplatin the references of 31 and 33 are missing. The text lines 160-169, 248-268 etc explain clearly the effect of vinblastin during pregnancy. Similar can be found in the text for other drugs.
Maybe the authors want to show in the table the effect of only one drug for example Vinblastin without the combination of any other drugs. If such choice is taken, I did not clearly understood the table. Such strategy should be clearly indicated.
Round 3
Reviewer 3 Report
No further comments